# Reinforcement Learning with Dynamic Movement Primitives for Obstacle Avoidance

Ang Li [1,2], Zhenze Liu [3], Wenrui Wang [1,2,*], Mingchao Zhu [1,*], Yanhui Li [1], Qi Huo [1] and Ming Dai [1]

[1] Changchun Institute of Optics, Fine Mechanics and Physics, Chinese Academy of Sciences, Changchun 130033, China; liang@ciomp.ac.cn (A.L.); liyanhui@ciomp.ac.cn (Y.L.); huoqi@ciomp.ac.cn (Q.H.); daim@ciomp.ac.cn (M.D.)
[2] University of Chinese Academy of Sciences, Beijing 100049, China
[3] College of Communication Engineering, Jilin University, Changchun 130025, China; zzliu@jlu.edu.cn
[*] Correspondence: wangwenrui16@mails.ucas.ac.cn (W.W.); zhumingchao@ciomp.ac.cn (M.Z.)

**Abstract:** Dynamic movement primitives (DMPs) are a robust framework for movement generation from demonstrations. This framework can be extended by adding a perturbing term to achieve obstacle avoidance without sacrificing stability. The additional term is usually constructed based on potential functions. Although different potentials are adopted to improve the performance of obstacle avoidance, the profiles of potentials are rarely incorporated into reinforcement learning (RL) framework. In this contribution, we present a RL based method to learn not only the profiles of potentials but also the shape parameters of a motion. The algorithm employed is PI2 (Policy Improvement with Path Integrals), a model-free, sampling-based learning method. By using the PI2, the profiles of potentials and the parameters of the DMPs are learned simultaneously; therefore, we can optimize obstacle avoidance while completing specified tasks. We validate the presented method in simulations and with a redundant robot arm in experiments.

**Keywords:** obstacle avoidance; Dynamic Movement Primitives; reinforcement learning; PI2 (policy improvement with path integrals)

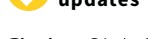



## 1. Introduction

As robots are applied to more and more complex scenarios, people set a higher request to adaptability and reliability at the motion planning level. To deal with dynamic environments, there are at least two different strategies to avoid collision for robots. One is global strategy [1,2], it is usually based on search processes and often computationally expensive and time-consuming [3], such that continuous fast trajectory modification based on sensory feedback are hard to accomplish. The other is local strategy, it is always fast to compute, but the computed trajectories are suboptimal. To this end, Dynamic Movement Primitives (DMPs) [4] are introduced as a versatile framework to solve this problem.

### 1.1. Related Work

In DMPs framework, the additional perturbing term is modified online based on feedback from the environment to achieve obstacle avoidance [5–8]. The perturbing term is usually designed as different artificial potential fields to get better performance [5]. The classical one is static potential, and the effect of the static field is only related to the distance between the end-effector and obstacles [9]. The dynamic potential field is most combined with the DMP method [5]. By using the dynamic potential field, robots can accomplish smoother avoidance movements because the potential depends on both the distance and the relative velocity between end-effector and obstacles [6,7]. The closed-form of harmonic potential function was presented to avoid the convex and concave obstacles [10]. It inherited the convergence of the harmonic potential while not resulting in the creation of a potentially infinite number of pseudo attractors on the obstacle [11].

Although different forms of potential have their own advantages in different obstacle avoidance scenes, it is very difficult to choose different potentials according to the scene. In addition, the scenarios that robots work in are not invariable, so it is necessary to introduce the framework of learning into obstacle avoidance.

It is possible to apply human beings learning skill to robot obstacle avoidance. In particular, the robot motion can be governed by a demonstration trajectory with DMPs. Then, in a similar way as human beings adjust their position in the process of obstacle avoidance, parameters of the potential function and DMPs can be adjusted through learning based on certain criteria. These kinds of learning approaches have been developed in a lot of research. A method was presented to learn the coupling term of DMPs from human demonstrations to make it more robust while avoiding a larger range of obstacles [12]. This algorithm has the ability to model different types of obstacle avoidance behaviors by learning coupling terms, and uses the learned coupling terms to avoid obstacles in a reactive manner. To improve the performance, a neural network was introduced to learn a function that predicts the coupling term given sensory inputs [13]. The neural networks prediction is based on physical constraints, which guaranteed that the obstacle avoidance behavior was always stable and converged to the goal position. A biologically-inspired hierarchical learning framework was also used to modulate the coupling term of DMP, which guides and regulates the obstacle avoidance behavior [14]. A multi-layer perception-decision-action analysis framework was adopted to extract unified low-dimensional geometric descriptors of system obstacles, and then use them in combination with heuristics and learning techniques for rapid reasoning of the environment. Most of the obstacle avoidance learning algorithms focus on the learning of the coupling terms, while rarely considering the correction of the overall DMP-encoded policy, especially the adjustment of the weight terms of the Gaussian function. DMPs could be optimized by learning algorithms like Relative Entropy Policy Search, by combining the approach with potential fields, and two proposed extensions significantly increase the probability that the joint angle is within the normal range in the reference trajectories and achieve obstacle avoidance without further sensors [15]. However, the potential field with fixed parameters is adopted here, which will not be flexible enough in the whole obstacle avoidance process.

In many scenarios, such as robot assembly, robot welding, and robot handling, DMP can help the robot avoid obstacles by collecting information about the surrounding space with the help of sensors. On the premise of ensuring the learning ability of DMP for the trajectory, improving the obstacle avoidance performance of the robot has important research significance. A learning framework is presented that incorporates DMP weights and learning coupling terms in this paper. One possible learning method to develop this framework is Reinforcement Learning (RL) [16]. A well designed reward function is significant to RL in obstacle avoidance. Final strategies with good performance would be determined by achieving a high reward score when the algorithm is trained repeatedly [17]. The employed RL method in this work is PI2 (Policy Improvement with Path Integrals), which has outstanding performance in optimizing DMPs [18–20]. PI2 can simultaneously optimize planned trajectories and obstacle avoidance potential in a DMP in this framework. In our framework, firstly, a DMP is initialized with an initial trajectory and pre-defined potential. Then, the DMP and potential function parameters are updated iteratively by executing PI2 to accomplish the desired behavior while avoiding obstacles. Meanwhile, the cost for each resulting trajectory is computed with the task-specific cost function in the process. During execution, exploration is ensured through adding exploration noise to the DMP and potential function parameters. Therefore, we can fully focus on the cost function design to optimize the DMP parameters and potential function and obtain a better performance in obstacle avoidance. We validate the performance of our approach on simulated scenarios including different shapes of obstacles and different dimensions of obstacle avoidance. We also validate the presented framework on a real 7-DOF redundant manipulator. The manipulator is required to track accurately through a via point while avoiding collision with obstacles on the demonstration trajectory.

*1.2. Contribution*

This article contributes to the following aspects:

1. The PI2 method is employed to optimize the planned trajectories and obstacle avoidance potential in a DMP;
2. A well designed reward function which combines instantaneous rewards and terminal rewards is proposed to make the algorithm achieve better performance;
3. Simulations and experiments on a real 7-DOF redundant manipulator are designed to validate the performance of our approach. In addition, a simulation with specified via-point shows the flexibility in trajectory learning.

The remainder of this paper is organized as follows: in Section 2, we recall the theory of obstacle avoidance for DMPs and analyze the effect of potential strength on obstacle avoidance. Section 3 discusses how to use the RL algorithm (PI2) to simultaneously optimize trajectory shape generated with DMPs and the potential field strength for obstacle avoidance. In Section 4, simulations and experiments are conducted by performing obstacle avoidance tasks in two and three dimensions to verify our approach. Thereafter, the conclusions are drawn in Section 5.

## 2. Obstacle Avoidance for Dynamic Movement Primitives

Dynamic motion primitive is a trajectory learning method that can modify its ongoing control strategy with a reactive strategy, so it can be used for obstacle avoidance. Now, we briefly review the formulation of DMPS and how to accomplish obstacle avoidance with DMPs.

The movement trajectory can be generated by using DMPs. The differential equations of DMPs are inspired from a modified linear spring-damper system with an external forcing term [21]:

$$\tau \dot{\mathbf{v}} = \mathbf{K}(\mathbf{x}_g - \mathbf{x}) - \mathbf{D}\mathbf{v} + (\mathbf{x}_g - \mathbf{x}_0)\mathbf{f}(t)\boldsymbol{\theta}, \tag{1}$$

$$\tau \dot{x} = v. \tag{2}$$

where $\mathbf{x}$ and $\mathbf{v}$ are, respectively, the displacement and velocity; $\mathbf{x}_g$ is the goal point of a movement while $\mathbf{x}_0$ is the starting point; $\mathbf{K}$ is the stiffness matrix corresponding to the stiffness of the system while $\mathbf{D}$ is the damping matrix; $\tau$ is a constant scaling factor to determine the movement period. The last term on the right-hand side of (1) is modulatable and parameter vector $\boldsymbol{\theta}$ can be learned to generate arbitrarily movements. $\mathbf{f}$ is Gaussian basis function, and it is defined as

$$(\mathbf{f}(t))_j = \frac{w_{j,t}s_t}{\sum_{k=1}^{p} w_{k,t}}. \tag{3}$$

In the definition equation for $\mathbf{f}(s)$, $w_j = \exp\left(-0.5h_j(s_j - c_j)^2\right)$ is Gaussian function with parameters $h_j$ and $c_j$ corresponding to the center and width. $\mathbf{f}(s)$ depends explicitly on a phase variable $s$ instead of time. $s$ is defined as

$$\tau \dot{s} = -\alpha s, \tag{4}$$

where $\alpha$ is a preset constant. (4) can be called a 'canonical system'. The state variable s with initial value 1 converges to 0 during the moving duration.

To achieve the avoidance behaviors, a repellent acceleration term $\boldsymbol{\varphi}(\mathbf{x}, \mathbf{v})$ is added to the transformation system (1)

$$\tau \dot{\mathbf{v}} = \mathbf{K}(\mathbf{x}_g - \mathbf{x}) - \mathbf{D}\mathbf{v} + (\mathbf{x}_g - \mathbf{x}_0)\mathbf{f}(s)\boldsymbol{\theta} + \boldsymbol{\varphi}(\mathbf{x}, \mathbf{v}). \tag{5}$$

For the additional term, one of the most commonly used forms is to model human obstacle avoidance behavior with a differential equation. Here, we will leave aside the concrete dimensions while only constructing a general form. It can be extended to high or

low dimensional space depending on the actual tasks. The differential equation is written as [5]

$$\boldsymbol{\varphi}(\mathbf{x}, \mathbf{v}) = -\nabla_{\mathbf{x}} \mathbf{U}(\mathbf{x}, \mathbf{v}) = \lambda \mathbf{R} \mathbf{v} \alpha \exp(-\beta \alpha), \tag{6}$$

where $\mathbf{U}$ is a potential depending on position $\mathbf{x}$ and velocity $\mathbf{v}$; $\beta$ and $\lambda$ are constants, $\lambda$ is the potential field intensity factor, and it can determine the strength of the repellent term in 2D or 3D space; $\mathbf{R}$ is a rotation matrix of angle $\frac{\pi}{2}$ with the axis $\mathbf{r} = (\mathbf{o} - \mathbf{x}) \times \mathbf{v}$, where $\mathbf{o}$ denotes the obstacle position and $\times$ denotes the cross product; $\alpha$ is a steering angle, and it is defined as [6] (shown in Figure 1).

$$\alpha = \arccos\left(\frac{(\mathbf{o} - \mathbf{x})^T \mathbf{v}}{\|\mathbf{o} - \mathbf{x}\| \|\mathbf{v}\|}\right), \tag{7}$$

where $\|\|$ is the Euclidean norm. The value of $\alpha$ is always positive. In this work, we only focus on this formulation of the repulsive term. Compared with other approaches, the formulation can guarantee convergence to the goal position, even if the effects of obstacles persist during the whole movement duration. Because the variables $\mathbf{x}$, $\mathbf{v}$ and $\mathbf{o}$ are all available, obstacle-avoidance movements are also easily obtained by using these extended DMPs.

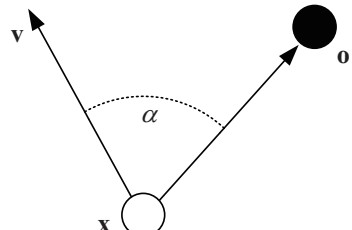

**Figure 1.** The diagram of the steering angle $\alpha$.

As we mentioned before, the strength of potential filed is largely determined by $\lambda$ and the effect of $\lambda$ is evident on movement generation. Therefore, different $\lambda$ lead to different behavior in obstacle avoidance. By setting different values of $\lambda$, there is different performance in obstacle avoidance as shown in Figure 2. It is not difficult to find that, if $\lambda$ is set to a smaller value such as $\lambda = 0.003$ (see the orange curve in Figure 2), the generated movement may fail in an obstacle avoidance; if $\lambda$ is set to a bigger value such as $\lambda = 0.1$ (see the purple curve in Figure 2), the movement has a bad performance in tracking the desired trajectory. Choosing an appropriate $\lambda$ is important for generating a better movement (see the green curve in Figure 2). In other words, it is very essential to choose a better potential field for obstacle avoidance and good tracking performance.

To our knowledge, the profiles of the generated movement with DMPs are determined not only by the obstacle avoidance repulsive term but also by the parametrized nonlinear term. To this end, if we want to obtain a trajectory with good performance in both obstacle avoidance and trajectory tracking, the parameters $\theta$ of the DMPs and the strength $\lambda$ of the potential are optimized simultaneously in the movement generation process. This could be regarded as a high-dimensional optimization problem obviously. Reinforcement learning algorithms are able to optimize motion primitive parameters efficiently and robustly in high-dimensional problems [19]. Thus, we will present a framework based on RL to optimize the parameter of DMP and the strength of the repulsive potential field simultaneously in the next section.

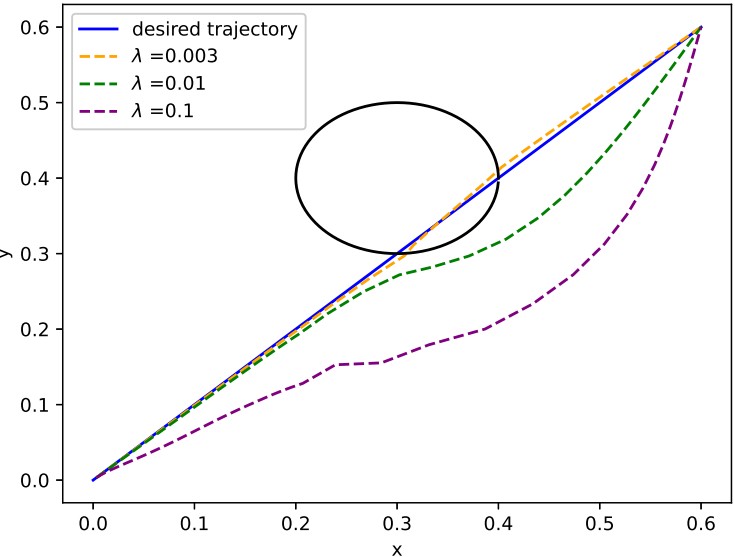

**Figure 2.** Obstacle avoidance behavior with different values of $\lambda$.

### 3. Reinforcement Learning for Obstacle Avoidance

*3.1. Reinforcement Learning: The PI2 Algorithm*

Autonomous learning systems are generally used in the field of control, and reinforcement learning is one of their frameworks [22]. PI2 (policy improvement with path integrals) is a scalable RL method for high dimensional continuous state systems. Moreover, PI2 is a sampling-based and model-free learning method, and algorithm parameter adjustment is not required. Therefore, we can fully focus on the cost function design in the process of applying PI2. The most important challenge is the cost function design in the application of RL algorithms including PI2. For PI2, the cost function is generally defined as

$$J(\boldsymbol{\tau}_i) = \phi_{t_N} + \int_{t_i}^{t_N} (r_t + \mathbf{u}_t^T \mathbf{R} \mathbf{u}_t) d_t, \tag{8}$$

where $J$ is the a finite horizon cost function for a trajectory $\boldsymbol{\tau}_i$; $\phi_{t_N}$ denotes the terminal cost at time $t_N$; $r_t + \mathbf{u}_t^T \mathbf{R} \mathbf{u}_t$ is the immediate cost at time $t$, and here, $r_t$ is an arbitrary state-dependent cost function, $\mathbf{u}$ is a policy parameter vector determined by $\boldsymbol{\theta}$ and $\lambda$, and $\mathbf{R}$ is a positive definite weight matrix. The goal of applying PI2 is to find the policy $\mathbf{u}_t$ that minimizes the cost function

$$V_t = \min_{\mathbf{u}_{t_i:t_N}} E_{\boldsymbol{\tau}_i}[J(\boldsymbol{\tau}_i)], \tag{9}$$

where the expectation $E_{\boldsymbol{\tau}_i}[.]$ is taken over all possible trajectories when the policy $\mathbf{u}$ is used.

In the process of applying the policy improvement method, we minimize the cost function through an iterative process of exploration and parameter updating. The principles of stochastic optimal control can be used to solve the PI2, and the details are discussed in [18]. The PI2 algorithm needs to be applied to the controlled dynamics systems with parameterized policy

$$\dot{\mathbf{x}}_t = \mathbf{f}_t + \mathbf{G}_t(\mathbf{u}_t + \boldsymbol{\varepsilon}_t), \tag{10}$$

where $\mathbf{x}_t$ is the system state; $\mathbf{f}_t$ is the passive dynamics, $\mathbf{G}_t$ is the control matrix, and they are both related to the system state; $\mathbf{u}_t$ is the control input with Gaussian noise $\boldsymbol{\varepsilon}_t$, The variance of the Gaussian noise is $\boldsymbol{\Sigma}$. The transformation system of DMP falls into one kind of this control system [22].

A second-order partial differential equation of value function is derived by minimizing the HJB (Hamilton–Jacobi–Bellman) equation of our problem

$$\partial_t V_t = r_t + (\nabla_\mathbf{x} V_t)^T \mathbf{f}_t - \frac{1}{2}(\nabla_\mathbf{x} V_t)^T \mathbf{G}_t \mathbf{R}^{-1} \mathbf{G}_t^T (\nabla_\mathbf{x} V_t) \\ + \frac{1}{2} trace\big((\nabla_{\mathbf{xx}} V_t)\mathbf{G}_t \mathbf{\Sigma} \mathbf{G}_t^T\big) \tag{11}$$

where $\nabla_\mathbf{x}$ and $\nabla_{\mathbf{xx}}$ are, respectively, the Jacobian and Hessian matrix. We can also obtain the optimal control input, and it is a function of the system state [17]

$$\mathbf{u}_{t_i}^* = -\mathbf{R}^{-1}\mathbf{G}_t^T(\nabla_{\mathbf{x}_{t_i}} V_{t_i}). \tag{12}$$

To solve the Equation (11), we use an exponential transformation $V_t = -\lambda \log \varphi_t$ and introduce an assumption $\lambda \mathbf{R}^{-1} = \mathbf{\Sigma}$. With this, we can get a linear second-order partial differential equation

$$-\partial_t \varphi_t = -\frac{1}{\lambda} r_t \varphi_t + \mathbf{f}_t^T(\nabla_\mathbf{x} \varphi_t) + \frac{1}{2} trace\big((\nabla_{\mathbf{xx}} \varphi_t)\mathbf{G}_t \mathbf{\Sigma} \mathbf{G}_t^T\big), \tag{13}$$

where a boundary condition should be satisfied that $\varphi_{t_N} = \exp\left(-\frac{1}{\lambda}\phi_{t_N}\right)$. According to the Feynman–Kac theorem [18,23], we can get the solution to the exponential transformation of the value function

$$\varphi_{t_i} = \int p(\boldsymbol{\tau}_i) \exp\left(-\frac{1}{\lambda}\left(\phi_{t_N} + \int_{t_i}^{t_N} r_{t_j} dt\right)\right) d\boldsymbol{\tau}_i, \tag{14}$$

where $p(\boldsymbol{\tau}_i)$ is the probability of a trajectory $\boldsymbol{\tau}_i$, and it can be written as:

$$p(\boldsymbol{\tau}_i) = \frac{e^{-\frac{1}{\lambda}s(\boldsymbol{\tau}_i)}}{\int e^{-\frac{1}{\lambda}s(\boldsymbol{\tau}_i)} d\boldsymbol{\tau}_i}, \tag{15}$$

where $s(\boldsymbol{\tau}_i)$ is defined as

$$s(\boldsymbol{\tau}_i) = \phi_{t_N} + \sum_{j=i}^{N-1} r_{t_j} dt + \frac{1}{2}\sum_{j=i}^{N-1} \left\| \frac{\mathbf{x}_{t_{j+1}} - \mathbf{x}_{t_j}}{dt} - \mathbf{f}_{t_j} \right\|_{\mathbf{H}_{t_j}}^2 d_t, \tag{16}$$

and $\mathbf{H}_{t_j}$ is expressed as

$$\mathbf{H}_{t_j} = \mathbf{G}_{t_j}\mathbf{R}^{-1}\mathbf{G}_{t_j}^T. \tag{17}$$

Thus, the optimal controls can be written in the expectation form

$$\mathbf{u}_{t_i} = \int p(\boldsymbol{\tau}_i)\mathbf{u}(\boldsymbol{\tau}_i) d\boldsymbol{\tau}_i = \int p(\boldsymbol{\tau}_i)\mathbf{R}^{-1}\mathbf{G}_{t_i}^T\big(\mathbf{G}_{t_i}\mathbf{R}^{-1}\mathbf{G}_{t_i}^T\big)^{-1}\mathbf{G}_{t_i}\boldsymbol{\varepsilon}_{t_i} d\boldsymbol{\tau}_i \tag{18}$$

In this way, the probability $p(\boldsymbol{\tau}_i)$ can be calculated by forward integrating the system dynamics and calculating the costs. Moreover, the optimal control policies can also be approximated by drawing random samples of the noise vector $\varepsilon$. The P-weighted sum of the local controls $u_{t_i}$ of these samples is used to approximate the value of the integral [17,18]. In the process of applying the PI2, the forward integrals of system dynamics are replaced by local controls extracted from probability distributions and running random controllers on the real system. We can get the cost statistics from the experiments. Each of the experiments is regarded as a roll-out.

### 3.2. Reinforcement Learning of Potential and Shape

PI2 is usually used to optimize the movement shape generated by DMP. It aims to minimize a cost function by tuning the policy parameters $\boldsymbol{\theta}$. Based on this, we extend the PI2 method to simultaneously optimize potential function parameters and movement

shape. According to (5) and (6), the transformation system with obstacle avoidance can be written as

$$\tau \dot{\mathbf{v}} = \mathbf{K}(\mathbf{x}_g - \mathbf{x}) - \mathbf{D}\mathbf{v} + (\mathbf{x}_g - \mathbf{x}_0)\mathbf{f}(s)\boldsymbol{\theta} + \lambda \mathbf{R}\mathbf{v}\alpha \exp(-\beta\alpha). \tag{19}$$

Since the PI2 algorithm is only a special case of optimal control solution, it can be applied to control systems with parameterized control policy [17]. By using the PI2 algorithm, the transformation system (19) can be parameterized

$$\tau \dot{\mathbf{v}} = \mathbf{K}(\mathbf{x}_g - \mathbf{x}) - \mathbf{D}\mathbf{v} + (\mathbf{x}_g - \mathbf{x}_0)\mathbf{f}(s)\left(\boldsymbol{\theta} + \boldsymbol{\varepsilon}^{\boldsymbol{\theta}}\right) + \left(\lambda + \varepsilon^{\lambda}\right)\mathbf{R}\mathbf{v}\alpha \exp(-\beta\alpha), \tag{20}$$

where $\boldsymbol{\varepsilon}^{\boldsymbol{\theta}}$ is the noise which is added to explore the movement shape and $\varepsilon^{\lambda}$ is added to explore the strength of the potential. In addition, $\boldsymbol{\varepsilon}^{\boldsymbol{\theta}}$ and $\varepsilon^{\lambda}$ are generated by sampling from Gaussian functions, and their variances are $\Sigma^{\boldsymbol{\theta}}$ and $\Sigma^{\lambda}$, respectively. Simultaneous optimization of these two parameters belongs to the category of hierarchical reinforcement learning [20]. The parameters $\theta$ and $\lambda$ are learned simultaneously on different levels of abstraction.

In the learning process, the exploration for the shape of DMP usually occurs in the fixed potential field. This means that the potential update should begin before updating the shape. Because the strength of potential $\lambda$ remains unchanged during execution, it is not temporal-dependent on the cost. To this end, only the cost-to-go at $t = 0$ is employed to calculate the probability of $\lambda$. The cost used is the total cost of the trajectory in the last roll-out. No negative interference exists between learning $\theta$ and $\lambda$, as they are updated with the equally probability weights because their costs are exactly equal [19,20]. The PI2 algorithm applied in learning both potential strength and DMP shape is summarized in Algorithm 1.

---

**Algorithm 1** PI2 algorithm for learning potential strength $\lambda$ and DMP shape parameters $\boldsymbol{\theta}$.

---

**Input:**
    Initial potential strength $\lambda_0$; initial DMP shape parameters $\boldsymbol{\theta}_0$; constant $\beta$; terminal cost $\phi_{t_N}$; immediate cost term $r_t + \boldsymbol{\theta}_t^T \mathbf{R}\boldsymbol{\theta}_t$; variance of noise $\boldsymbol{\Sigma}^{\boldsymbol{\theta}}$ and $\Sigma^{\lambda}$; Gaussian basis function from the system dynamics $\mathbf{g}_{t_i}$; number of roll-outs per update $K$

**Output:** Final potential strength $\lambda$; final DMP shape parameters $\boldsymbol{\theta}$

1: **while** cost $J$ not converged **do**
2:   **for** k = 1 to K **do**
3:     $\boldsymbol{\tau}_{k,i=0:N}, \boldsymbol{\theta}_{k,i=0:N}, \lambda_k = \text{create\_trajectory}\left(k, \boldsymbol{\theta}, \boldsymbol{\Sigma}^{\boldsymbol{\theta}}, \lambda, \Sigma^{\lambda}\right)$
4:     calculate generalized trajectory cost $s(\boldsymbol{\tau}_{i,k})$ with (16)
5:     calculate probability of a trajectory $p(\boldsymbol{\tau}_{i,k})$ with (15)
6:   **end for**
7:   **for** i = 0 to N **do**
8:     $\delta\boldsymbol{\theta}_{t_i} = \sum_{k=1}^{K}\left[p(\boldsymbol{\tau}_{i,k})\mathbf{M}_{t_i,k}\boldsymbol{\varepsilon}_{t_i,k}^{\boldsymbol{\theta}}\right]$, where $\mathbf{M}_{t_i,k} = \dfrac{\mathbf{R}^{-1}\mathbf{g}_{t_i,k}\mathbf{g}_{t_i,k}^T}{\mathbf{g}_{t_i,k}^T\mathbf{R}\mathbf{g}_{t_i,k}}$
9:     $[\delta\boldsymbol{\theta}]_j = \dfrac{\sum_{i=0}^{N-1}(N-i)\left[\delta\boldsymbol{\theta}_{t_i}\right]_j w_{j,t_i}}{\sum_{i=0}^{N-1}(N-1)w_{j,t_i}}$
10:   $\boldsymbol{\theta} = \boldsymbol{\theta} + \delta\boldsymbol{\theta}$
11:   **end for**
12:   $\mathbf{p}(\boldsymbol{\tau}_{0,k}) = \dfrac{e^{-\frac{1}{\lambda}s(\boldsymbol{\tau}_{0,k})}}{\sum_{l=1}^{K}\left[e^{-\frac{1}{\lambda}s(\boldsymbol{\tau}_{0,l})}\right]}$
13:   $\delta\lambda = \sum\limits_{k=1}^{K}\left[\mathbf{p}(\boldsymbol{\tau}_{0,k})\varepsilon_k^{\lambda}\right]$
14:   $\lambda = \lambda + \delta\lambda$
15: **end while**

---

Since the state of a DMP system can be divided into the controlled part and the uncontrolled part, in the meantime, the control transition matrix depends on only one variable of the uncontrolled part [18,21], and the trajectory cost of DMP can be given as

$$s(\boldsymbol{\tau}_{i,k}) = \phi_{t_N,k} + \frac{1}{2}\sum_{j=i}^{N-1} r_{t_j,k} + \frac{1}{2}\sum_{j=i+1}^{N-1} \left(\boldsymbol{\theta} + \mathbf{M}_{t_j,k}\boldsymbol{\varepsilon}_{t_j,k}^{\boldsymbol{\theta}}\right)^T \mathbf{R}\left(\boldsymbol{\theta} + \mathbf{M}_{t_j,k}\boldsymbol{\varepsilon}_{t_j,k}^{\boldsymbol{\theta}}\right), \quad (21)$$

where $\mathbf{M}_{t_j,k}$ is computable (cf. Algorithm 1). In this way, only local sampling and an iterative procedure need to be considered in the process of updating $\boldsymbol{\theta}$.

## 4. Simulations and Experiments

In this section, we will evaluate the algorithm for obstacle avoidance in simulations and experiments. The obstacles in our evaluations are modeled by using point clouds on the boundary [6]. The details of edge point detection are not discussed further here, the data of these points are available by default in our simulations and experiments. Thus, the positions of the edge points will be directly used in 2D or 3D.

The goal of our work is to achieve obstacle avoidance and get a good following of the desired trajectory. Therefore, we design the cost function for this task as

$$J(\boldsymbol{\tau}_i) = \phi_{t_N} + \int_{t_i}^{t_N} \left( G\left\|\mathbf{x}_t - \mathbf{x}_t^d\right\| + \frac{1}{2}\boldsymbol{\theta}_t^T \mathbf{R}\boldsymbol{\theta}_t \right) dt, \quad (22)$$

where $\phi_{t_N} = L(1 - successofavoidance)$. This means that, if the obstacle avoidance is successful $\phi_{t_N} = 0$; otherwise, $\phi_{t_N} = 1$, and we can set $L$ to a large value such as $L = 10,000$ and try to avoid collisions; $G$ is a constant, and we can select it based on the specific tasks; $\mathbf{x}_t^d$ is the desired position at time $t$; $\mathbf{R}$ is an identify matrix, and it is set as $\mathbf{R} = 10^{-5}\mathbf{I}$. The parameters for the PI2 learning algorithm, including the exploration noise for shape and potential strength, should be tailored respectively to individual context in the following simulations and experiments.

### 4.1. Simulations

In the first simulation, we will test and compare the behaviors in Section 2 and, using our RL algorithm in Section 3, performing the same tasks. In addition, in this simulation, we set $G = 10$, $\lambda_0 = 0.1$, $\beta = 1$, $K = 400$, $\boldsymbol{\Sigma}^{\boldsymbol{\theta}} = 0.01\mathbf{I}$ and $\Sigma^{\lambda} = 0.001$. It should be noted that $\boldsymbol{\theta}_0$ is learned from the initial desired trajectory. For the exploration noise $\boldsymbol{\Sigma}^{\boldsymbol{\theta}}$ and $\Sigma^{\lambda}$, higher exploration noise usually leads to quicker convergence, but it can also cause safety issues. The fixed noise is used in our work, so relatively small noise is chosen. In this way, the rate of convergence is slow; however, a better performance can be obtained. The learning results of potential field strength $\lambda$ are presented in Figure 3. The cost does not converge to a perfectly good value. In the meantime, the potential strength $\lambda$ converges to $\lambda = 0.0051$ in Figure 4. In order to evaluate the performance improvement of the method more intuitively, we calculate the sum of the distances(SOD) of all track points relative to the original ones in a DMP. We have calculated separately when SOD takes different values of $\lambda$. As shown in Table 1, potential strength calculated by PI2 performances is 10.4% better than a constant value 0.01.

**Table 1.** SOD with different values of $\lambda$.

| $\lambda$ | 0.0051 (PI2) | 0.01 |
|---|---|---|
| SD (cm) | 73.55 | 82.08 |

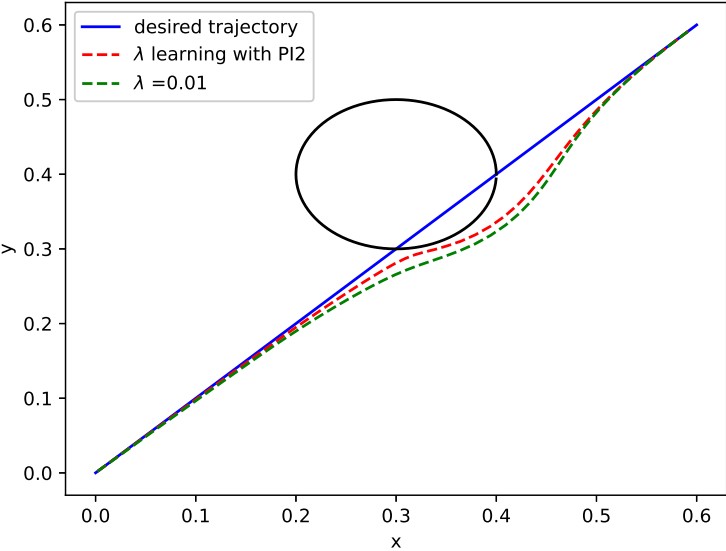

**Figure 3.** Obstacle avoidance behavior with RL algorithm: PI2.

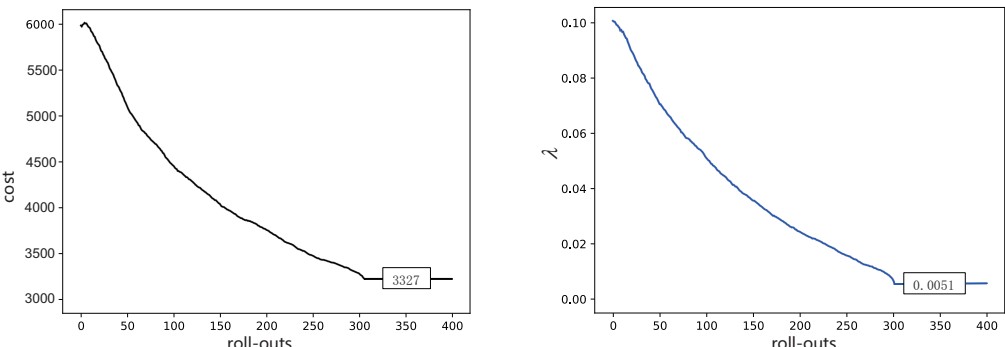

**Figure 4.** Learning curve for obstacle avoidance. Left: cost profiles with roll-outs. Right: the potential strength $\lambda$ profiles with roll-outs.

The PI2 algorithm used in this work is a random strategy improvement algorithm, but our optimization function focuses on the optimization of the overall trajectory, so it is difficult to achieve a particularly good overall effect under the condition of ensuring safety. To this end, we set a convergence threshold on the basis of selecting a suitable $K$ value to improve the convergence speed. After all, convergence to the global optimum cannot be guaranteed [22]. Even so, the advantages of PI2 algorithm are still very obvious, and we can design different cost functions to achieve obstacle avoidance while completing different sub-tasks such as via-point task. In the via-point task, the goal is to pass through the pre-set point $\mathbf{x}_{set} = \begin{bmatrix} 0.15 & 0.2 \end{bmatrix}^T$ at $t = 200$ ms, so the cost function of this task could be described as

$$J(\boldsymbol{\tau}_i) = \phi_{t_N} + \int_{t_i}^{t_N} \left( G \left\| \mathbf{x}_t^{200ms} - \mathbf{x}_{set}^{200ms} \right\| + \frac{1}{2} \boldsymbol{\theta}_t^T \mathbf{R} \boldsymbol{\theta}_t \right) dt. \tag{23}$$

In this cost function, we set $G = 10{,}000$, $K = 100$, $\boldsymbol{\Sigma}^{\boldsymbol{\theta}} = 0.1\mathbf{I}$ and $\lambda = 0.0051$, the optimized potential strength; other parameters are the same as the first simulation. However, different from the first simulation, this simulation focuses on illustrating shape learning, and show the relationship between costs over motion and the corresponding trajectory shape. Therefore, the updating of potential field strength is not considered here, but the optimized results from the first simulation are used as the parameter set. The results are shown in Figure 5. We can see that the initial curve with the optimized potential is the same as the first simulation. After about 60 updates shown in Figure 6, the costs for

via-point learning sessions have converged which terminates the training after that. For the via point learning with PI2, as the cost decreases from 19,996 to 20, the trajectory passes through the pre-set point (the blue point in Figure 5) without nearly any error. However, the performance for tracking the original desired trajectory is very poor while passing near the preset point. This is because it is far from the original desired trajectory both in time and space to let the trajectory pass through the pre-set point at $t = 200$ ms. Moreover, in the process of learning DMP shape $\theta$, $G$ is selected as a larger value, so it will naturally have a greater influence on the original trajectory shape and better pass through the preset target point. After passing the pre-set point, the trajectory will follow the original obstacle avoidance track to the maximum extent.

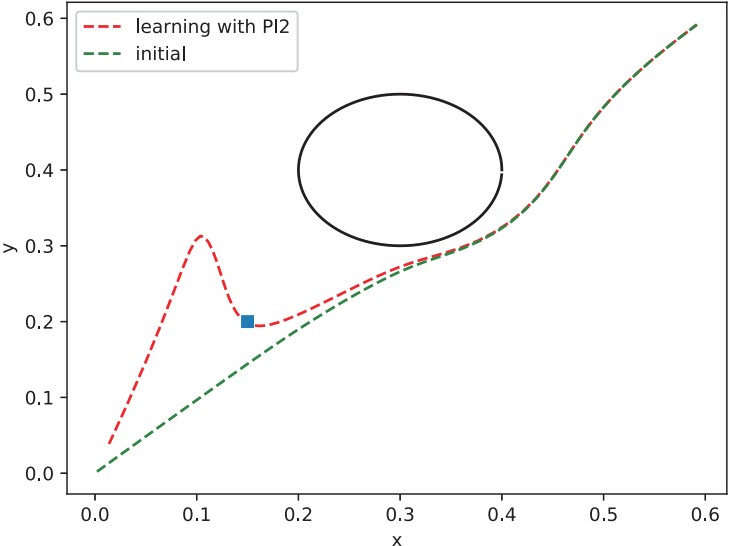

**Figure 5.** Obstacle avoidance behavior with different values of $\lambda$.

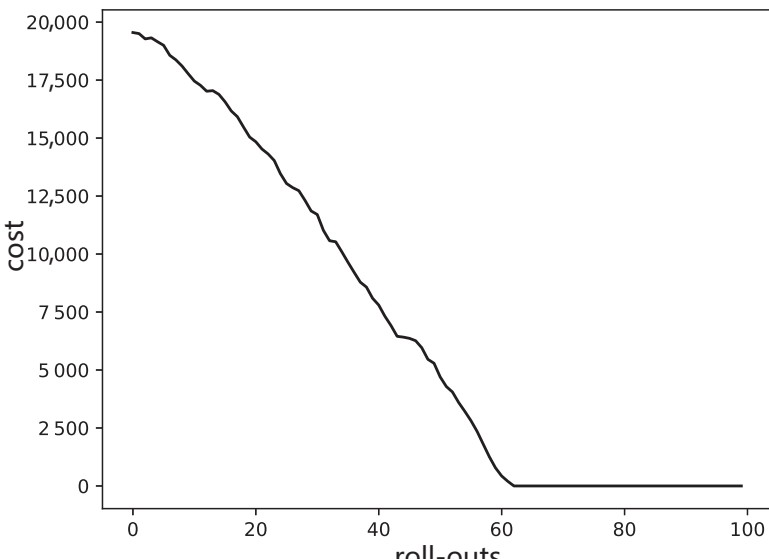

**Figure 6.** Via-point learning cost profiles with roll-outs.

In these two simulations, we consider two sets of learning situations. The first one is to simultaneously optimize obstacle avoidance and tracking effect of the desired trajectory. However, according to the results, the optimization effect of DMP shape is not obvious, but the potential field intensity can be optimized to a certain extent. The potential field strength optimized by our method can learn a better potential and get a better obstacle

avoidance performance. The second simulation is based on the optimized potential field strength, and we set another via-point target and modify the cost function. The simulation results are good and cost converges to a very small value.

### 4.2. Experiments

The goal of this task is for the real 7-DOF robot to track the trajectory learned from the demonstration, avoiding collision with an obstacle in the meantime. In addition, the RL method is used to optimize the performance in the task.

In the demonstration process, we pulled the end-effector of the robot according to the planned trajectory and the poses of the end-effector will be recorded over time. Here, we focus on trajectory and obstacle avoidance of the robot end-effector, and joint angles are solved automatically using inverse kinematics of the robot. The demonstrated trajectory in end-effector space is shown in Figure 7. We use a DMP with ten basis functions to train each pose of the end-effector. It should be noted that we are mainly concerned with the change in the translational pose position of the end-effector, so the rotational pose is fixed [3], which means only three working DMPs are learning the Cartesian positions of end-effector of the 7-DOF robot. The initial potential strength is 0.1, and the cost is designed as (22); only $G$ is changed to $G = 1$. The variance of the exploration noise of each pose is respectively 0.01, 0.001, and 0.01. The acquired point cloud of obstacle is processed by applying standard filtering techniques [14]. In addition, for the sake of simplicity, we are only going to consider the case where the obstacle is a sphere.

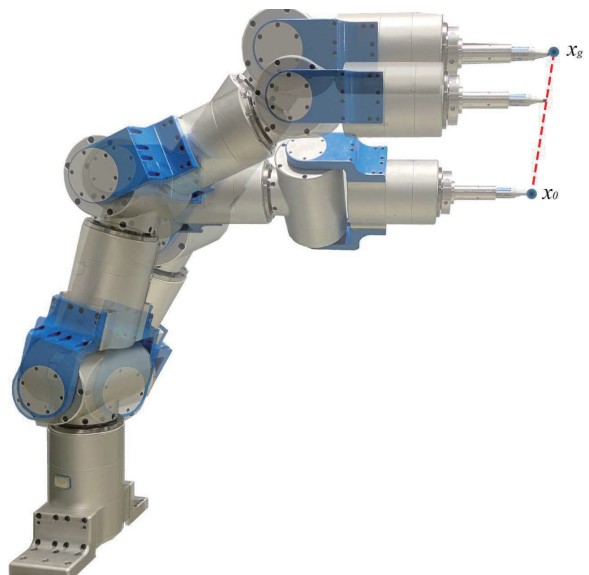

**Figure 7.** Obstacle avoidance behavior with different values of $\lambda$.

Figure 8 depicts the trajectories of a real experiment for the 7-DOF robot, the blue line is the desired trajectory (learned from demonstration), the red dotted line is the obstacle avoidance trajectory with constant value of potential $\lambda$ and trajectory shape $\theta$, and the green dotted line is the learned trajectory with PI2. The training and learning process is the same as the simulations, so this will not be described in more detail again in this paper. Compared with the training trajectories before and after the learning, it is obvious that the potential field strength and trajectory shape obtained by reinforcement learning method can make the real robot perform better in obstacle avoidance. This is consistent with the simulation results. Although the robot only performs the trajectory learned offline, the training information is projected into the training space using obstacles and trajectory information in the real space. If trajectory tracking error is not taken into account, the proposed reinforcement learning framework is available in most cases; furthermore, the RL framework can be extended to more practical applications.

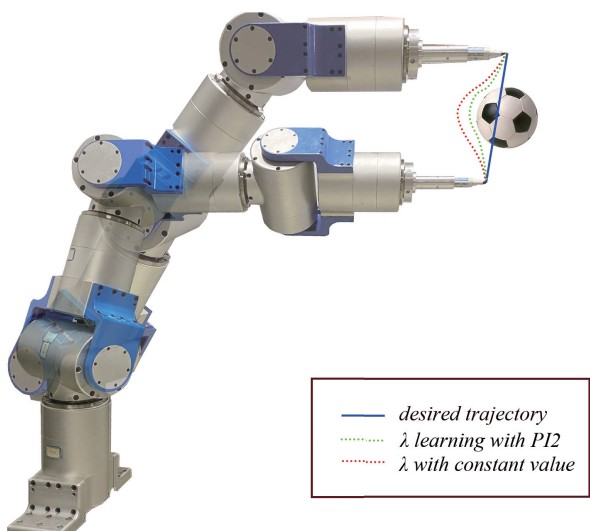

**Figure 8.** Obstacle avoidance behavior with different values of $\lambda$.

In summary, simultaneous learning potential and trajectory shape are available by using the prosed RL framework whether in simulations or real experiments. In addition, it enables the robot to obtain better performance in obstacle avoidance, tracking the desired trajectory and performing other subtasks.

## 5. Conclusions

In this paper, we propose a reinforcement learning framework for obstacle avoidance with DMP. The strength of repulsive potential is incorporated in the RL framework, such that the shape of DMP and the potential are optimized simultaneously. Because the RL algorithm PI2 is a model-free, probabilistic learning method, different task goals can be achieved only by designing cost functions. To optimize obstacle avoidance performance, we pick the overall tracking error as cost function, and set a large terminal cost in the case of obstacle avoidance failure. The proposed approach is evaluated in 2D obstacle avoidance. The potential strength is optimized and the tracking is improved to some extent. PI2 is a suboptimal stochastic optimization method; therefore, many more attempts are necessary if you want to achieve better performance. Even so, it is verified that simultaneous learning of potential and shape is valid in the proposed RL framework. In addition, then, we test our RL framework by adding a sub-task, via-point. In this situation, it can not only maintain good obstacle avoidance performance but also can successfully achieve passing through the pre-set point. We also evaluate the approach on one 7-DOF robot, and the evaluation demonstrates that the algorithm behaves as expected in real robots.

**Author Contributions:** Conceptualization, A.L. and W.W.; methodology, A.L. and W.W.; software, A.L., W.W. and Z.L.; validation, A.L., W.W. and Y.L.; formal analysis, A.L., W.W. and Q.H.; investigation, W.W.; resources, M.Z.; data curation, A.L. and W.W.; writing—original draft preparation, A.L. and W.W.; writing—review and editing, A.L.; visualization, A.L.; supervision, W.W.; project administration, M.Z. and M.D.; funding acquisition, M.Z. All authors have read and agreed to the published version of the manuscript.

**Funding:** This research was funded by project "Fire Assay Automation" of Changchun Institute of Optics, Fine Mechanics and Physics, Chinese Academy of Sciences.

**Institutional Review Board Statement:** Not applicable.

**Informed Consent Statement:** Not applicable.

**Data Availability Statement:** The data presented in this study are available on request from the corresponding author W.W. The data are not publicly available due to the data also forming part of an ongoing study.

**Acknowledgments:** The authors are grateful to the Science and Technology Development Plan of Jilin province (2018020102GX) and Jilin Province and the Chinese Academy of Sciences cooperation in the science and technology high-tech industrialization special funds project (2018SYHZ0004).

**Conflicts of Interest:** The authors declare no conflict of interest.

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
