# Peer review of "Reinforcement Learning with Dynamic Movement Primitives for Obstacle Avoidance"

_applsci, doi:10.3390/app112311184_

Round 1

Reviewer 1 Report

The theme of paper is good and this paper is well written on reinforcement learning with dynamic movement primitives for obstacle avoidance. The authors validate the presented method in simulations and with a redundant robot arm in experiments. The following are some typos errors:

Page 1, line 13: Insert right bracket ")" at the end.

Page 3, line 103: What is "thes"?

Page 7, line 221: What is "te"?

Page 10, line 270: What is "forst"?

Page 10, line 297: "figure 7"  should be "Figure 7"

Page 11, line 303: Check, is there any one 0.1? in the data 0.01, 0.001, 0.01 

Page 11, line 319: "... is available.." should be "... are available.."

Page 11, line 331: "the potential ..." should be "The potential ..."

Page 12, lines 350-356: Delete extra text "Authors must .............. publish the results.""

Pages 12-13, in references: (i) Write all names instead et. al.
      (ii) Use one kind of abbv in all references, for example,  either H or H. for short name.

Author Response

Point1: Page 1, line 13: Insert right bracket ")" at the end.

Response 1:The right bracket ")" has been inserted.

Point2 Page 1, line 13: Page 3, line 103: What is "thes"?

Response 2: “thes” has been replaced by“the” .

Point3: Page 7, line 221: What is "te"?

Response 3: “te” has been replaced by“the”  .

Point4: Page 10, line 270: What is "forst"?

Response 4: “forst” has been replaced by“first” 

Point5: Page 10, line 297: "figure 7"  should be "Figure 7"

Response 5: “figure” has been replaced by“Figure” 

Point6: Page 11, line 303: Check, is there any one 0.1? in the data 0.01, 0.001, 0.01 

Response 6: These three numbers are used to describe the variance of the exploration noise in the three directions of XYZ in Cartesian space during training of the reinforcement learning algorithm. They are not initialization environmental parameters in different scenarios, so they can produce the same value. Their values have been verified and there is no problem.

Point7: Page 11, line 319: "... is available.." should be "... are available.

Response 7: “is” has been replaced by“are”

Point8: Page 11, line 331: "the potential ..." should be "The potential ..."

Response 8: “the” has been replaced by“The”

Point9: Page 12, lines 350-356: Delete extra text "Authors must .............. publish the results.""

Response 9: This piece of content has been deleted

Point10: Pages 12-13, in references: (i) Write all names instead et. al.
      (ii) Use one kind of abbv in all references, for example,  either H or H. for short name. Response 10: (i) All names have been added to the reference information

(ii) The format of abbreviations has been unified into a capital letter with “.”

Reviewer 2 Report

Thank you for inviting me to be a reviewer of the manuscript entitled Reinforcement Learning Using Dynamic Movement Primitives for Obstacle Avoidance. This paper is really impressive in terms of your efforts to demonstrate the power of your method.

I suggest you change the structure of the paper and add a research methodology with research questions and hypotheses. I also suggest adding and clarifying the research objectives. I also suggest expanding the discussion of the results and findings. 

Round 2

Reviewer 2 Report

The corrected version of the article is much better.